# Do You Feel Safe at Home? A Qualitative Study among Home-Dwelling Older Adults with Advanced Incurable Cancer

**DOI:** 10.3390/healthcare10122384

**Published:** 2022-11-28

**Authors:** Ellen Karine Grov, Siri Ytrehus

**Affiliations:** 1Department of Nursing and Health Promotion, Faculty of Health Sciences, Oslo Metropolitan University, 0130 Oslo, Norway; 2Department of Health and Caring Sciences, Western University of Applied Sciences, 5020 Bergen, Norway

**Keywords:** cancer care, older adults, mental health, feeling safe, home-dwelling

## Abstract

Many older adults with cancer prefer to live at home, and home treatment and outpatient care have been recommended for such patients. To improve their mental health, it is important to identify the challenges that are faced by home-dwelling older adults with cancer. This study aimed to examine the impact of the home on older adults with advanced cancer who were receiving treatment and follow-up care. In a cross-sectional design with criterion-based sampling, eight qualitative interviews were transcribed and interpreted thematically. We identified three themes of home-safety management: good home-safety management, uncertain home-safety management, and home-safety management collapse. Moreover, we revealed eight sub-themes important to the participants’ home-safety experience. Ensuring that older adults feel safe at home will afford them the opportunity to enjoy living at home, which in turn may alleviate their symptom burden and enhance their mental health.

## 1. Introduction

As the World Health Organization states, mental health “…goes beyond the mere absence of a mental health condition” and includes, e.g., social, cultural, and other interrelated factors contributing to mental health [1]. With this broad definition, we assume safety and reduced risk of harm as aspects of mental health on both system and individual levels. Thus, with support from a Finnish study, we argue that mental well-being and dwelling in a safe environment are linked together [2]. Over the past few decades, health authorities have underscored the role that the home plays in the lives of older adults with advanced health problems and the need for home care and social services [3,4,5,6]. The relationships that older adults share with their homes and communities have been gaining renewed research attention as a result of the increase in the number of older adults and the commonness of a combination of out-patient treatment services and home-based care and treatment. Many older adults continue to live in their own homes even when they have advanced cancer, receive advanced treatment, and follow-up palliative care [7,8,9]. 

Currently, in many countries, health care service providers assume that home-based care is more cost effective than institutional care [10], and it is assumed that older adults wish to continue living in their homes for as long as possible [11]. However, some of these assumptions have been questioned [11,12]. 

### 1.1. The Meaning of a Home to Older Adults

Across the fields of social geography, sociology, and gerontology, the dominating perspective on homes is that the home should be understood not only as a physical space within which various events take place but also as an entity that is influenced by both emotional, mental, and social factors [13,14]. Different conceptualisations of a home have been documented in the literature [13]. Research studies on older adults and their homes have focused on the meaning that older adults ascribe to the physical, emotional, mental, and social dimensions of a home. The importance of the social and emotional dimensions of a home have been repeatedly underscored, and safety, which is one aspect of these dimensions, has been regarded as a central component of attachment between a person and his or her home [15]. The feeling of safety that a home engenders is emphasised as a fundamental sense of safety or ontological security that is embedded within the material surroundings [16]. 

### 1.2. The Home and Older Adults with Cancer 

Relatively little is known about the assistance that older adults with cancer need [17,18,19], their experience of having advanced incurable cancer while living at home [20], and the effects of treatment on their mental and social functioning [21,22]. Similarly, little is known about the meanings that home-dwelling older adults with advanced cancer assign to a home whilst living alone. It can be very difficult to live alone under such circumstances [9,17,19,23]. Older adults are likely to be dependent on others (who often do not belong to the household) for all of their activities. Living alone can exacerbate their sense of uncertainty about the future [23,24]. Older adults with cancer who live in rural areas are a particularly vulnerable group [19,25,26,27]. Long distances to hospitals can limit their access to requisite help [28,29]. Furthermore, they have to travel afar to gain access to treatment services and health institutions. Norwegian studies, which were conducted among home-dwelling older adults with cancer in rural areas, found that older adults consider waiting for transportation and uncomfortable post-treatment transportation to their homes to be sources of great distress [19,25]. 

Palliative care research has focused on the therapeutic role that close physical surroundings in in-patient settings play during this final phase of life [30,31]. Palliative care research studies have shown that older adults prefer to die in their own homes [30,32,33]. However, this contention has been challenged because the pertinent studies have not adequately accounted for the alternatives that are available to them. There have only been a few attempts to examine safety initiatives within the home care sector [34,35]. Within a patient’s home, his or her safety is inextricably linked to that of his or her carer [34]. Trusting staff, being comfortable and well informed, and engaging in daily life activities contribute to well-being and a sense of security among those who receive palliative home care [35,36]. However, there is a lack of research findings on the dimensions of a home that contribute to a sense of safety. 

Therefore, this study aimed to identify the dimensions of a home that older adults consider to be important and examine the role that advanced cancer plays in the meanings that older adults with this disease assign to their homes and their feelings of safety and mental well-being when living at home. The resultant findings can be used to develop strategies that promote the safety of home-dwelling older adults with serious health problems and accommodate their extensive needs for assistance. We have outlined the following questions for this study: what role does the home play for the older adults living at home with an advanced cancer disease, what role does the disease play in the meaning of the home, and which dimensions of the home are highlighted as significant?

## 2. Methodology

Qualitative research methods facilitate the identification of the nuances of the relationships that individuals share with spaces [37,38]. Therefore, a qualitative cross-sectional research design was adopted in this study with criterion sampling, and we found thematic analysis suitable for data navigation [39].

### 2.1. Recruitment Strategy and Sample

In this study, participants were recruited in collaboration with nurses in the home care service representing four municipalities. Older adults who met the inclusion criteria received documents that (a) requested their participation in the study, (b) provided information about the study, and (c) required them to provide informed consent. Their responses to the request were relayed to the researcher. Men and women who were older than 60 years and had advanced incurable cancer were eligible for inclusion. The exclusion criteria were as follows: individuals with a cognitive impairment or a condition (e.g., terminally ill patients) that interferes with their ability to participate in an interview. A total of eight older adults from the four municipalities agreed to participate in this study. This study comprises of municipalities with 10,000–25,000 inhabitants, defined by us as rural or sub-urban. 

The ages of the participants ranged from 60 to 86 years (Table 1). Five of them were men, and three of them were women. Three of them lived alone, and the others lived with a partner or spouse. They all lived in their own private homes. The participants had the following different types of cancer: breast, colorectal, prostate, bladder, oesophageal, and oesophageal and tracheal cancer. One participant had more than one type of cancer. All participants demonstrated poor functioning and had major health problems, which were caused by cancer, but the consequences of cancer and health problems varied across participants. We had no access to the participants’ medical or nursing record. With these limitations, we have no objective assessment as Karnofsky or ECOG performance status. Thus, we have described their health conditions in the same way that they told us and based on our assessment. Both authors have practiced as nurses for several decades. All the participants received public or private assistance. 

### 2.2. Interviews 

A semi-structured interview guide that covered specific topics, which were identified from the literature, was developed. The interviews were conducted in the participants’ homes. The participants reported that they felt relaxed during the interview, and they were given the opportunity to show us their homes. In the first interview, both authors served as active interviewers. The intention was to harmonise the way the interviews were conducted, and the data were collected. Regarding the other interviews, the first and last author conducted four and three interviews, respectively. The interviews were conducted with the assistance of the interview guide. The following are a few sample questions: can you describe what it is like to live at home when you have cancer? What does being able to live at home mean to you? What kind of help do you receive, and what are your experiences of receiving such help? Follow-up questions were used to facilitate further reflection upon and exploration of participants’ experiences. Such questions were: how do you deal with daily activities at home? Do you find any tasks challenging? Please tell about making meals. Follow-up questions on shopping and cooking were conducted. What does this house mean to you? Please describe what is preferable, if so, of staying at home despite your serious illness. Are you able to walk outdoors? Do you need help from others to make the bed and clean the house? Who are your helpers? Are you receiving support or help from the home care nursing service?

Each interview lasted for 70 to 120 min. The spouses or partners of the participants participated in four interviews. They participated because the older adults wanted them to. 

All the interviews were recorded and transcribed. The names and locations of the participants were deidentified during transcription. In this article, we present parts of the interviews, where ‘I’ refers to the interviewer and A-H are letters representing the patients’ voice. We have connected the letters A-H to the quotations, a system that corresponds to the tables.

### 2.3. Analysis

Thematic analysis was used [39] whereby the data were analysed from quotations through sub-themes and themes ending with an overarching theme. Firstly, we became familiar with the data, a process which had already started while interviewing the participants. We separately read the transcripts and the noted text parts (coding units) and then in face-to-face meetings, we discussed our understanding of what we interpreted as the essential meaning. The authors always had the research questions in mind (the home’s role for the patient, and the disease’s impact on the meaning of the home). No software was used for the thematic analysis, as the authors wanted to read the text several times and wanted closeness to the data. From the text parts (coding unites), we identified central quotations and from these, we suggested and discussed sub-themes, themes, and the overarching theme. In the dialogue, we viewed the overarching theme, the themes and the sub-themes the other way—by turning back to the quotations. In Table 2, we present descriptions of ‘good home-safety management’; ‘uncertain home-safety management’ and ‘home-safety management collapse’. Each of the sub-themes have connected quotations in the text and we have used headings corresponding to the sub-themes and themes in the Results session. 

However, we have exceeded the analysis to show our interpretation of each of the participants’ positions on home-safety management. In Table 3, we provide a schematic overview of our analysis presented in four phases. In phase 1, we systematically examined and reviewed the transcripts and coded the data by critically analysing the text. We found descriptions of the home as a point of departure, where the dimension of the home included the environment, the house and the role that cancer played in the participants’ lives as well as the life situation with the availability of a partner, family or neighbours.

In phase 2, we shed light on what impacts each participant experienced, and a person-centred perspective from the participants (A–H) and contextual analysis [38] was conducted to examine the significance of various dimensions of the home in relation to each participant’s overall life situation. In this analytic phase, we directed our attention towards the sub-themes (referred in Table 3) that had emerged from the interview data of each participant. The aspects of the participants’ life situations were embedded within the interview data, and we therefore classified this phase with dimensions of the participants’ life situation. Phase 3 focuses on the analytical approach, which allowed us to understand the meanings that the older adults with advanced cancer ascribed to a home. The disease is present for the participants, and we have labelled this phase ‘the disease’s role in the meaning of the home’. In phase 4, the themes of home-safety management were described, and the characteristics of the participants’ home-safety management were identified. The matrix presented in Table 3 summarises the phases of this analytical process. 

### 2.4. Ethics

The Regional Committees for Medical and Health Research Ethics (no.1524) and the chief health officer in each municipality approved this study. The data were stored in a secure location during the study and will be moved to the Norwegian Sikt (https://sikt.no/) (accessed on 24 October 2022).

Data services and storage (giving data access in the Norwegian language) will be conducted after the completion of the study. The participants provided informed consent, and they were informed about their right to withdraw from the study at any time.

## 3. Results 

### 3.1. Good Home-Safety Management

This theme comprises of the informants’ descriptions on their wish to stay at home and their feeling of being safe at home. Several participants wanted to continue living in their own homes, despite their extensive health issues and uncertainty about the progress of their disease. They wished to maintain the home as it was and reported that they thrived when they were at home and experienced well-being there. 

### 3.2. Valuable Characteristics of the Homes—Predictability and House Quality 

The health issues of some participants made them particularly aware of the value of their home. Two male participants expressed their gratitude for being able to continue living in their homes. The disease had forced them to give up engaging in a variety of activities. They reported that their homes afforded them the opportunity to spend time outdoors. One of them emphasised the satisfaction, mental well-being, and pride that he derived from living in a single-family dwelling and being able to walk in the garden that surrounded his house. He reported that this was something that he did not take for granted, given his high age and his advanced disease.


*C: Well, yes, that’s how it is. We’re lucky to be alive; we’re the oldest people around here, more or less. Oh, we are just so grateful for this!*



*I: Yes.*



*C: We’re really doing fine. We can walk outdoors, you know.*


One of the other participants in this theme had advanced metastatic cancer and needed assistance with most daily tasks, including mobility. The disease prevented him from travelling or going for walks. The activities that he engaged in in his home were a positive compensation for the activities that he had to give up. 


*A: We have not gone any further than the terrace and the deck. But it’s amazing how great that can be too.*



*I: Yes.*



*A: We’ve said a few times, this summer, that this hotel here is certainly a pretty good place to live.*


### 3.3. Caregiver Availability and Close Relationship 

Both these men lived with a partner who played a central role in their current lives. The partners actively participated in the interviews. The men emphasised that their female partners had taken on substantial and necessary responsibilities for their care. Both stated that they would not have been able to live at home without their partners. They both said that they felt safe and wanted to continue living in their current home.

One of the female participants (E) expressed a strong desire to continue living in her home. Her partner also participated in the interview. She lived in an old apartment building that was located at the centre of a mid-size municipality. Her living conditions were significantly poorer than those of the other participants. It was difficult to access the apartment because it had a steep staircase. She had to exit the apartment to use the bathroom and toilet. Nevertheless, she strongly emphasised that she wished to continue living in her home. The municipality had offered her a new and better-adapted house in an assisted living facility. She and her partner had declined this offer. Her partner was better informed about the disease and treatment than she was. Indeed, he reported that he was responsible for contacting the health services and he was involved in performing other household tasks. However, his contributions were different from those of the other partners. Specifically, neither he nor the woman explicitly emphasised that his contribution played a decisive role in her ability to continue living in her home. The participant repeated several times during the interview that she was independent.


*E: “I want to stay here. He wants to stay here. This is our home!”*


### 3.4. Activities, Memories, and Independent Life in the Home

A third male participant in this theme of good home-safety management had advanced prostate cancer. He lived alone in a single-family dwelling with a garden in a rural area. He had always lived alone. He faced many challenges as a result of his cancer. He reported a series of problems, such as the following, for which it had been difficult to receive help: pain, problems with mobility, incontinence, and uncertainties about changes in and the progress of the disease. Nevertheless, he emphasised that he thrived, felt safe at home, and was able to make his own decisions. Furthermore, he was surrounded by familiar objects and lived in an environment that he had a part in creating. He also spoke about his interest in interior design. 


*I: You have said a bit about wanting to live at home. What does that mean to you?*



*B: Quite a lot.*



*I: Could you say a bit more about this? In which way?*



*B: I feel good here.*



*I: What is it that makes you feel good here?*



*B: My surroundings—they’re just what I want.*



*I: How’s that?*



*B: Furniture, curtains. I sew it all myself. I sew curtains for people too.*


He wished to continue living in his home. His stories about his exhausting treatment journeys made it apparent that he had taken great efforts to continue living in his home. 


*B: What was exhausting was the bus drive with the Health Express from X (located far away from the specialist hospital) and down to XX (specialist hospital) and back. It took six hours.*



*I: Yes.*



*B: It was exhausting, but I had to get home to my own bed and my cat.*


### 3.5. Uncertain Home-Safety Management 

In this theme, we revealed a lot of uncertainty. This was expressed in terms of concerns about the home being the right dwelling place and the feeling of being insecure at home and uncertainty about the future because of the disease’s progression. In some individuals, the disease created a sense of uncertainty about whether they ought to move. This was the case for participants who lived alone in single-family dwellings. They did not know what sort of assistance they would need in the future and how the disease would progress; therefore, they considered moving. However, both participants expressed that they felt safe at home at the time of the interview. Thus, they had considered moving because the disease had created uncertainty about the future, not because they were dissatisfied with their home or regarded it as being unsafe. They no longer took their homes for granted.

### 3.6. Independent but Unprotected

One participant considered a range of alternative places to which he could move, but he had not decided about what he wanted to do. His current home was important to him. This made it difficult for him to move. He also emphasised the following factors: his independence, the substantial effort that he had invested in maintaining his home, the fact that it had a terrace, and his emotional attachment to his home.


*H: There was an advertisement where they wanted new tenants. I thought, ‘This would be good for me’. I went up to look at it. It’s nice. It is one of those terraced house things. There was assistance twenty-four hours a day. I could get food served.*



*I: Yes.*



*H: But then, I sit here at night. I have, after all, spent some time on building this conservatory.*



*I: It’s really nice.*



*H: I just really love this place here.*


### 3.7. Feeling Insecure about Living Alone

When participant H spoke about his feelings of uncertainty and mentioned that he had been looking for a new home, he added:


*H: I was a bit scared then. You know, it is a bit difficult for me to get out of this place. I just had no intention of selling; I thought I would rent. But, if I leave here. I couldn’t do it.*


The female participant (F) described that she was dependent on her husband, and without him she could not cope with her situation at home. Two cancer diagnoses, dizziness and balance problems resulted in her story about feeling insecure in a scenario living alone.


*F: My husband helps with all the practical tasks at home.*


### 3.8. Home-Safety Management Collapse

The informants defined to be in the ‘home-safety management collapse’ theme expressed challenges regarding their life situation and disease progression. They wanted to move because they felt unsafe at home. 

### 3.9. Spending Parts of the Day Alone in the Home

Only two of the eight participants reported that they felt unsafe at home. Both spent considerable parts of their day alone. In both the participants, cancer had resulted in difficulties with mobility and pain. One participant was a woman who was living in a terraced house with her partner. Her partner worked full time and was away during the day. The other participant was a man who lived alone in an apartment on the first floor of a single-family dwelling. His adult son and his partner lived on the ground floor. 


*D: My husband is at work the whole day.*


### 3.10. High Symptom Burden

The female participant had breast cancer with pelvic and brain metastasis. She reported a long-term disease trajectory, many hospital stays, and long periods where she had endured great difficulties in receiving the necessary help from health care services, especially when the symptoms of the metastasis of breast cancer had emerged. At the time of the interview, her disease status was stable. She had experienced several fractures and undergone several operations (including surgery to stabilise the joints) as a result of her cancer. She had fallen and was injured several times. She reported pain and several symptoms, which can reasonably be attributed to metastasis to the brain. She reported that she felt very unsafe when she was alone at home. 


*D: …back home, here, afterwards, you know, and being alone, when my husband went to work. At the hospital, they would rather get rid of me. They were looking for a place where I could go to stay, and yeah, get some training or something.*


After a long process, she was informed that she could stay in a palliative care unit for a few days. She found this solution to be satisfactory. 

The other participant, categorized here, was in great pain. He had comorbidities (cancer and diabetes) and complications due to his diseases:


*G: All my toes on one of my feet are amputated.*


### 3.11. Unable to Move Around 

The latter mentioned male participant who reported feeling unsafe at home expressed a strong desire to move and permanently live in a nursing home. He lived in the same house in which his son was living. His son was away during the day. The participant spent most of his day passively sitting in a chair. He explained that he wished to move because he felt unsafe when he was alone at home and faced challenges when he performed daily tasks. He referred to a respite stay that he had got access to in a nursing home. There he had felt safe and had received the help that he needed. 


*G: I am not able to move around. I am stuck here in this chair.*


## 4. Discussion

In this study, we examined how older adults with advanced cancer experience their homes, identified the dimensions of their homes that they consider to be important for their home-safety, and investigated the role that cancer plays in the meanings that they assign to their homes. The overarching theme of home-safety management is outlined with three themes.

Theories on late modern societies regard the home as a place that offers shelter and safety. The role that homes play in fostering a sense of security and mental well-being is particularly important because of constant changes in and the occasional unpredictability of societal conditions [16]. Advanced cancer causes unpredictable changes to an individual’s abilities, mental health, and experience of their life situation. Accordingly, the need to identify the factors that determine whether older adults feel safe in their own homes has been previously underscored [40]. From this study, we have learned that older adults enjoy living in their homes, and their homes had positive dimensions that enhanced their daily lives. The significant dimensions of the home that were emphasised by the participants were pleasure from the features of their homes, having opportunities to engage in activities, being independent, taking pride in their homes, being grateful for the opportunity to be outdoors, and having access to a garden. Thus, homes can alleviate the negative consequences of cancer and optimize well-being. In such contexts, it is important to conceptualise the relationship between the disease, the mental health status, and the home as a dynamic whole. Specifically, home-safety can relegate cancer to one’s mental status, and cancer in turn can make older adults particularly aware of the various valued aspects of their homes. In this regard, the home may serve as a stimulus and represent something positive.

From an older adult’s perspective, feelings of safety are an intermediary dimension and condition. For a positive synergy to exist between an older adult and his or her home, he or she must feel safe. The possibility of leading the life that one wishes to live can help older adults feel safe in their own homes [41]. According to Giddens (1991), a sense of safety is related to unconscious psychological aspects. By the overarching theme of home-safety management with its themes and sub-themes, we articulate the role that serious illnesses such as advanced cancer play in one’s sense of safety. A sense of safety, which may otherwise be an unconscious and taken-for-granted feeling state, can emerge as an issue that older adults consciously reflect upon and address. Partners play an essential role in this preparatory process. Sharing their homes with their families or partner helped the participants maintain their home, enjoy it, and feel safe in their homes. In this regard, the partners of those with several health issues and those who required substantial assistance made substantial contributions. Their home-safety was maintained because of the contributions of their partners. Concordantly, Aspell et al. (2019) [3] found that the intensity of home support was a predictor of admission to long-term care among home-dwelling older adults. Another study found that older adults in nursing homes feel safer than their home-dwelling counterparts do; this finding is attributable to the easier access to health care personnel [40]. In our study, some participants who lived alone wanted to continue living in their homes, despite facing many difficulties and having health issues. Despite having an advanced disease and a great need for assistance, some older adults felt safe living alone in their own homes.

Decades of research have shown that caring for seriously ill home-dwelling older adults is a great burden to their families [42]. In our study, the partners of some participants had assumed responsibility for home-safety management. Home-safety management can be experienced in different ways and to different extents and can foster a sense of perceived control in older adults. Other studies have shown that maintaining some level of control over one’s life when moving into a nursing home influences one’s sense of safety [43]. Some older adults recognised and placed substantial emphasis on their dependence on their partner. However, some older adults did not recognise their need for help and dependence, and their partner’s contributions were not explicitly acknowledged. Although the partners were responsible for home-safety management, the participants had not explicitly handed over the responsibility to their partners. This may have affected how both parties experienced this situation and contributed to their burden. In accordance with Wahl and Lang’s (2006) description of attachment between a person and his or her home, the present findings delineate specific social and emotional dimensions.

Unsurprisingly, some felt uncertain about continuing to live at home. They did not know how their disease could progress and how their life situations might change in the future. When older adults live alone and become dependent on others, uncertainty regarding their home-safety management levels may arise. However, they may not know when and how dependence will happen and what kind of help they will be offered. Thus, he or she is in a moratorium. To a certain degree, they may be aware of their life situation and may think of possible solutions. Several studies have highlighted the particular challenges that are faced by home-dwelling older adults in rural areas who have to travel long distances to receive health care and social services [19,28].

Feeling safe is a precondition for a sense of meaning, mental well-being, and enjoyment of one’s home. By conceptualising the home as a space that fosters meaning, safety, and routinised practice, we acknowledge that perceptions of homes are influenced by innumerable experiences that can change in the future [30,44]. Thus, a home has different meanings to different older cancer patients, and these meanings can change with time. A sense of safety and the possibility of maintaining a routinised practice may diminish, depending on personal and household resources and resources in the surroundings. From a recently published article, the authors emphasize discharge planning for end-stage cancer patients [45]. We assume that the same promoting and barrier factors as ethical considerations, and self-efficacy and experience among nurses are also relevant for other countries such as Japan, where discharge planning was studied. A study from Scandinavia shows that good collaboration across health systems [7] has positive effects on the perceived safety of home-dwelling older adults. In our study, some participants felt unsafe even though they shared their homes with their partner or family. This was attributable to prior experiences of difficulties in getting the necessary help, difficulties coping on a daily basis, mobility issues, a fear of falling, and a general sense of uncertainty. The extent of help those older adults require plays a significant role in their sense of safety; severe health issues can reduce their sense of control and, consequently, perceived safety [40]. Thus, receiving proper treatment and care does not guarantee a sense of safety [46].

By examining whether older adults perceive their homes to be spaces within which they feel safe or insecure and unsafe, health care and social services can ascertain the perceived home-safety management levels of older adults. When home-safety management is emphasised, health personnel and social workers should further assess the options that are available to older adults, particularly those with uncertain and collapsed home-safety managements.

The strength of this study is the data collected from a group of home-dwelling older adults with advanced cancer living in different areas in the country (rural and sub-urban municipalities). We did not aim to arrive at generalisable findings. Nevertheless, we acknowledge that using a larger sample may have yielded different results. The sample was sufficiently heterogeneous to facilitate the identification of the significant characteristics of home-safety management. Both male and female older adults with different marital status participated in this study. The participants had extensive and complex health problems. However, we are aware of the difficulties in defining patients to various stages of advanced cancer, as the palliative phase might also include the terminal phase [47]. Based on ethical considerations, we wanted to exclude those most vulnerable cancer patients, defined by us to be in the terminal phase. Through the thematic analysis, the authors engaged in a valuable dialogue, seeking interpretations to understand the data. A limitation in this study is the sample, where patient groups, such as those from ethnic minorities, were not included. The participants were not asked to provide feedback on the results. However, transparent description of the reflexive analytical process should enhance the study’s trustworthiness [48]. No software was used for the thematic analysis, and NVivo or other software could have organised the data in a different way than what we did manually.

This study might serve in revealing the generation of hypotheses for further studies. We suggest that studies with large samples should be conducted and studies including urban-dwelling participants.

## 5. Conclusions

Home-safety management assessment could be an approach to obtain an overview of how home-dwelling older adults with advanced cancer describe their homes, life situation and how the disease impacts their meaning of their home. It is important to adopt a holistic approach to care and pay attention to individual differences in the feelings of being unsafe, which can arise from various situations. We have elaborated on a description of home-safety management and outlined three themes: good home-safety management, uncertain home-safety management, and home-safety management collapse. Modifying the home environment to foster a sense of home-safety may give older adults with advanced cancer the opportunity to experience mental well-being and enjoy living in their homes. This may alleviate the burden of living with cancer.

## Figures and Tables

**Table 1 healthcare-10-02384-t001:** Description of participants by gender, age, living arrangements, disease and health issues, public health services, and help from family/partner.

Gender, Age, Living Arrangements	Disease/Function/ Health Issues	Assistance	Help from Partner/Family
A. Male, 74 years old, lives with wife.	Lung cancer with metastasis to colon. Leakage from colon following surgery. Breathing problems and poor mobility due to hip problems. Was able to go out to his own car until a few months ago. Can barely walk up and down stairs. Previous heart attack and thrombosis. Uses several medicines.	Private home services.	Son helps with shopping.
B. Male, 78 years old, lives alone.	Prostate cancer with metastasis to bones. Chemotherapy, no curative intention. Out-patient treatment. Very weak during weekly chemotherapy. Severe pain and uses morphine-based pain relief plasters. Weak and tired, sleeps poorly, and anxiety issues. Confusion, but he attributes this to sleep medication. COPD. Has venous access port (VAP), and urinary catheter.	Receives treatment from various hospitals: national cancer hospital, central hospital, and local hospital. Some private home health services.	Some help from neighbours.
C. Male, 82 years old, lives with wife.	Bone marrow cancer, back fracture, operated to stabilise joints. In a lot of pain, reduced vision, and swollen legs. Chemotherapy and morphine-based plasters. Eating problems. Needs help to get out of bed and help to move around.	Home care nurse assists with medication administration.	Wife helps moving around indoors.
D. Female, 60 years old, lives with husband.	Breast cancer with pelvic and head metastasis. Somewhat disoriented due to the metastasis to the head. Coordination problems. Weight loss during treatment. Previous hip fracture still bothering her. Unable to go out on her own.	Home care nurse. Days in a palliative unit. Followed up by primary physician.	Husband helps with all practical tasks at home. He also helps his wife moving indoors and outdoors.
E. Female, 71 years old, lives with partner.	Oesophageal and tracheal cancer, heavy breathing, heart problems, diabetes, and several wounds. Enteral nutrition only. Difficulties walking stairs. Unable to go out on her own. Difficulties sleeping, prescribed sleeping pills.	Help from home care nurse several times a week. Follow-up by primary physician.	Husband deals with enteral nutrition and administration of medication.
F. Female, 68 years old, lives with her husband.	Breast cancer and colorectal cancer with colostomy. Special diet, significant weight loss. Tired and weak, becomes dizzy, and balance issues. Difficulties sleeping. Still receiving chemotherapy as tablets. Changes bandages every day. Can go out on her own.	Colostomy nurse weekly.	Husband helps with all practical tasks at home.
G. Male, 80 years old, lives alone, son has apartment in same house.	Cancer of the bladder, urostomy, diabetes, and all toes on one leg amputated. Has received radiation treatment. In a lot of pain. Physical therapy. Goes out on his own.	Home care nurse daily. Needs help with caring for fistula, administration of medication, and showering.	
H. Male, 86 years, lives alone old, partner stops by occasionally.	Prostate cancer, in wheel-chair. Has stair lift and many aids installed in home. Impaired mobility, but the kitchen is well modified and he has a telephone on a table right next to where the wheelchair is placed.	Home care nurse daily. Needs help with showering and shopping	No help from family, but partner stops by occasionally. Considered moving.

**Table 2 healthcare-10-02384-t002:** The overarching theme ‘home-safe management’ with themes and sub-themes.

Home-Safety Management		
Good Home-Safety Management	Uncertain Home-Safety Management	Home-Safety Management Collapse
Valuable characteristics of the homes—predictability and house quality Caregiver availability and close relationship Activities, memories, and independent life in the home	Independent but unprotected Feeling insecure about living at home	Spending part of the days alone in the home High symptom burden Unable to move around

**Table 3 healthcare-10-02384-t003:** Schematic overview of analysis phases and the characteristics of the participants and their connected home-safety management.

	Phase 1	Phase 2	Phase 3	Phase 4
	Home dimension	Dimensions of the participant’s life situation	The disease’s role in the meaning of the home	Characteristics of home-safety management
Participant: A C E	Possibilities for going outdoors on the terrace and the balcony. spaciousness, takes pride in the home. Lives with partner.	The disease affects daily life to a great extent. Partner’s contribution is extensive. The partner’s contribution is both explicit and implicit. Emphasises meaningful and valued qualities of the home.	Enjoys the home. The disease underscores the meaning of the home. At the same time, the home places the disease in the background.	Good home-safety management. Home-safety management partly coped with by the participant or taken over by caregivers or relatives.
Participant: B	Activities, memories, interior, takes pride in the home. Lives alone.	The disease affects daily life to a great extent. Makes great personal efforts to remain in the home. Emphasises own efforts in forming the home.	Enjoys the home. The disease underscores the meaning of the home. At the same time, the home places the disease in the background.	Good home-safety management. Independent safety management.
Participant: F H	Takes pride in the home, garden, terrace, emotional attachment to the home, wants independence. One lives alone, one with a partner.	The disease necessitates extra efforts to remain in the home, but everyday life is, in the main, the same as before. Emphasises meaningful and valuable qualities of the home and own efforts in forming the home.	Enjoys the home. Considers moving but is uncertain. The disease creates uncertainty regarding the home in the future.	Uncertain home-safety management.
Participant: D G	Emphasises no aspect of the home. Lives with partner/family.	The disease affects daily life to a great extent. The participant is passive and immobile, cannot go out without help, difficulties moving around, has periodically not received needed help.	Does not wish to be alone in the home. Expresses no enjoyment of the home. The disease makes the participants unable to enjoy the home.	Home-safety management collapse.

## Data Availability

The data will be stored at the SIKT, previously the Norwegian Social Science Data Services (NSD), after completing the study. https://sikt.no/about-sikt (accessed on 24 October 2022).

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
