# Peer review of "Do You Feel Safe at Home? A Qualitative Study among Home-Dwelling Older Adults with Advanced Incurable Cancer"

_healthcare, 2022, doi:10.3390/healthcare10122384_

Round 1

Reviewer 1 Report

By the means of explorative qualitative research the authors tried to develop a model of safety management in three (?) dimensions (exact labels?: home features, engage in activities, independency, pride in home, outdoors opportunities, access to garden) and three levels (safe, insecure, unsafe), for home dwelling older persons with progressed cancer in four Norwegian municipalities (n=8, plus 4 spouses).

Although this safety approach is novel, and for an interesting target group, I have are major concerns with this paper, the methods and the clearity and conceptualisation of the safety model. The structure and labeling of dimensions (how many? labels?) and levels (three?) stay unclear throughout the whole paper and confusing in its wording (starting right from the Abstract where dimensions seem to be explained in terms of levels; lines 17-20). The methodological coding steps are vague and not explained in a reproduceable manner. The sample (8 older persons in 4 municipalities, and suddenly also 4 spouses, not coupled to the 8 older persons -could be done in tabel 1- and not discussed in chapter 4) is very small  and in no way representative within Norway (rural/urban; line 391? ) or outside Norway (context of health care system, variation in supply of services?) and no methodological rigour nor reflections/discussion about saturation (crucial requirement for qualitative research). Coding of quotes (with Xs) not aligned to codes A-H in table 1. Selection of precise quotes and whole interview fragments is not very convincing and not very illustrative for the content of the results and other results cry for a quote (for example in 3.1.2,  3.2.1, 3.3.1 (what is the real result here, we only read descriptives about respondents), and 3.3.3). Some results have not been presented in chapter 3, but pop up in the Discussion (example: lines 346-349).

The Discussion is quite lengthy and not very concise, although the embedding in literature is the better part of it. The claim that the strength of the paper is the rich data is far too strong. Clearity about saturation (not only heterogenity) could have helped to make a claim about conceptual in stead of statistical generalisablety. Could you add recommendations for both future research and future practice of home construction and health care support that incorporates your safety management model?

Author Response

Three documents attached:

1) Comments to the reviewer

2) Manuscript with tracked changes

3) Clean, readable manuscript

Reviewer 2 Report

Review for Manuscript entitled “Do you feel safe at home? A qualitative study among home-dwelling older adults with advanced incurable cancer”.

Thank you for providing me with the opportunity to review the manuscript entitled “Do you feel safe at home? A qualitative study among home-dwelling older adults with advanced incurable cáncer”

1.      Introduction:

Congratulations, Introduction is written clearly.

2.      Methodology

The methodological explanation of this qualitative study is very poor.

-What sampling strategy has been followed for the recruitment of participants?purposive sampling? Snowball sampling? Criterion sampling? It is necessary to specify this methodological aspect in order to carry out a qualitative study. Why was it decided to make this sample number? Has data saturation been reached?

The standards established by this author should be reviewed: https://doi.org/10.1080/13814788.2017.1375091

-As it was mentioned in the introduction, it would be interesting to mention whether the patients were in rural or urban environments. Since in other publications it is mentioned that home-dwelling older adults with cancer in rural areas can generate added stress.

-“In this study, participants were recruited in collaboration with four municipalities.” Although the names and locations of the patients were anonymous during transcription, it would be interesting to know the municipalities or provinces of the patients recruited. 

It is proposed for further review since it is necessary to specify the basic characteristics that regulate qualitative studies.

Author Response

(The authors gave the same response as above.)

Reviewer 3 Report

Thank you for the opportunity to review this manuscript. It is an important subject to provide voice to those living with advance cancer and being at home.

I have some edits, comments, and suggestions that can be taken into account: 

-abstract, line 1-2. Line 2 introduces mental health. If this is the focus of the paper, would recommend introducing this in line 1 which has no mention of this. 

-abstract: "model of safety management" can mean many things. I think it is meant in the home environment, consider inserting the word home before safety

-line 48- what has been documented? Consider adding to sentence: ....including...(brief description)

-"advanced cancer"- consider defining this terminology. Is this based on stage of cancer? functional status? symptom burden? undergoing cancer-directed treatment? 

-noted exclusion of "terminally ill patients." I would think many of these patients mentioned with advanced cancer are terminally ill. I wonder if it should read serious illness that affects retention of decision-making capacity

-Table 1: consider making it clearer and consistent. Include medical history, symptom burden, functional status, polypharmacy (can define). "Spreading" of cancer could be called metastasis. In the assistance column, noted that 2 participants had follow-up by PCP. Did others have this or other follow-up? I would think so? 

-line 118 what is in the semi-structured interview guide? consider including in supplement and refer to it. Line 125 then includes a "few sample questions"

-line 121- mentioned both authors participated in the interview, was this the first interview that was done together? if so mention it as the first as line 122 indicates desire to harmonize. 

-in qualitative methodology, there is mention of rigor and trustworthiness of the data. This is not incorporated into this manuscript. 

-line 141- "more inductive" is used. Was the other part deductive? 

-2.4 Ethics- consider moving this section above analysis section 

-3. consider introducing the results as a way to frame the next part of the manuscript. Are there themes, categories, domains? How were things coded? In what format? Data bases? Who coded it? What if there were discrepancies? 

-in the interview guide, was there a question about the reality that with advanced cancer, one may not be able to stay in their home and how they felt? 

-line 354- consider reviews of resources or lack of resources available to maintain home 

-line 383 "model of safety management was developed"-- likely need a more comprehensive discussion on its own. Is this model new? does it align with research out there? What is the future plans with this new model? 

-line 395 "different civil statuses" what is meant by this? 

-line 396- what does certain level of mental functioning mean? Does this refer to underlying decision making capacity, carrying a depression or anxiety diagnosis? This is not specified and not mentioned anywhere else in the paper. 

Author Response

(The authors gave the same response as above.)

Round 2

Reviewer 1 Report

The authors did a good job revising their paper.

After final round of spelling check, this paper seems publishable to me.

Author Response

Thank you!

Reviewer 2 Report

Review for Manuscript entitled “Do you feel safe at home? A qualitative study among home-dwelling older adults with advanced incurable cancer”.

Thank you for having made the modifications that the 3 reviewers have requested to the manuscript entitled “Do you feel safe at home? A qualitative study among home-dwelling older adults with advanced incurable cáncer”

However, with respect to my review, I feel that there are aspects that have not been solved. As I said in the previous review, it is necessary, within the methodology, to specify a series of terms that are key for qualitative research. It is necessary to specify, in order to be able to classify, for example, what type of sampling is: snowball sampling, criterion sampling, typical case sampling...

The methodology section has hardly been modified.

Again, I share with you the publication where you can be guided:

https://doi.org/10.1080/13814788.2017.1375091

I recommend that you follow the tables in the previous publication as they are very informative and allow us to classify the qualitative study. For example, I attach one of the tables where the type of sampling is specified.

In addition, you refer that the modifications made for my review are from page 9 to page 11. It is true that necessary aspects have been clarified but they are aspects related to the results. However, the methodological aspects that I mention to you should be defined and mentioned in the methodology section.

IT IS NECESSARY TO SPECIFY THE METHODOLOGICAL ASPECTS IN ORDER TO ACCEPT THIS PUBLICATION BECAUSE IT IS MANDATORY TO SPECIFY THE TYPE OF STUDY, STUDY DESIGN, ...

Author Response

Please find the attachment titled: Healthcare_202073_rev_2.

Reviewer 3 Report

Kudos to the authors for making revisions by reviewers. 

Author Response

Thank you!